# Using unstructured crowd-sourced data to evaluate urban tolerance of terrestrial native animal species within a California Mega-City

Joseph N. Curti[1]*, Michelle Barton[2], Rhay G. Flores[1], Maren Lechner[1], Alison Lipman[1], Graham A. Montgomery[1], Albert Y. Park[1], Kirstin Rochel[2], Morgan W. Tingley[1]*

**1** Department of Ecology and Evolutionary Biology, University of California, Los Angeles, CA, United States of America, **2** LA Sanitation and Environment, Los Angeles City, CA, United States of America

* jcurti3@g.ucla.edu (JNC); mtingley@g.ucla.edu (MWT)

**Data Availability Statement:** All relevant data and code are made available via a Zenodo repository (https://zenodo.org/records/10807319).

## Abstract

In response to biodiversity loss and biotic community homogenization in urbanized landscapes, there are increasing efforts to conserve and increase biodiversity within urban areas. Accordingly, around the world, previously extirpated species are (re)colonizing and otherwise infiltrating urban landscapes, while other species are disappearing from these landscapes. Tracking the occurrence of traditionally urban intolerant species and loss of traditionally urban tolerant species should be a management goal of urban areas, but we generally lack tools to study this phenomenon. To address this gap, we first used species' occurrences from iNaturalist, a large collaborative dataset of species observations, to calculate an urban association index (UAI) for 967 native animal species that occur in the city of Los Angeles. On average, the occurrence of native species was negatively associated with our composite measure of urban intensity, with the exception of snails and slugs, which instead occur more frequently in areas of increased urban intensity. Next, we assessed 8,348 0.25 x 0.25 mile grids across the City of Los Angeles to determine the average grid-level UAI scores (i.e., a summary of the UAIs present in a grid cell, which we term Community Urban Tolerance Index or CUTI). We found that areas of higher urban intensity host more urban tolerant species, but also that taxonomic groups differ in their aggregate tolerance of urban areas, and that spatial patterns of tolerance vary between groups. The framework established here has been designed to be iteratively reevaluated by city managers of Los Angeles in order to track the progress of initiatives to preserve and encourage urban biodiversity, but can be rescaled to sample different regions within the city or different cities altogether to provide a valuable tool for city managers globally.

## Introduction

The Earth is experiencing an extinction crisis, with modern species extinction rates, based on vertebrate taxa, estimated to exceed background rates of extinction by at least an order of magnitude [1,2]. In this contemporary era of species loss, there are a multitude of factors driving global declines including habitat loss, invasive species, disease, direct exploitation, pollution,

**Funding:** The author(s) received no specific funding for this work.

**Competing interests:** The authors have declared that no competing interests exist.

and human-caused climate change [3–7]. Many of the effects of these extinction drivers are increased due to synergistic interactions [8]; particularly, urbanization is well known to compound all of these drivers of extinction [8–10]. Globally, urban cover is predicted to increase by 2.5% between 2000 and 2030, such that urbanization will continue to increase as a driver of biodiversity loss [11]. Increases in urban cover are predicted to grow especially fast within global biodiversity hotspots, potentially by 200% between 2000 and 2030 [11], which could further exacerbate rates of species decline. With this predicted increase in urban areas, city managers and conservation biologists can work collaboratively to make cities more hospitable to native biodiversity in order to help avert increasing levels of extinctions.

The impacts of urbanization on biodiversity are well documented, especially in birds [12], arthropods [13,14], and plants [15]. Research on urban biodiversity has traditionally focused on quantifying changes in species richness along the urban-rural gradient (e.g., [16]). For example, studies demonstrate that native woodland bird species tend to be replaced by urban-adapted species in more urbanized habitats nearer to the city center [17,18], and that this pattern appears to be stronger for migratory species compared to residents [19]. Patterns of bird diversity are also tied to factors such as vegetation composition and structure. For example, canopy cover and native plant species diversity correlate with increases in native bird species richness in urban woodlands [20–22]; whereas, increases in lawn cover are related to increases in non-native and synanthropic species richness [23]. In other taxonomic groups, climatic variation has been shown to modulate the strength of the relationship between urbanization and species diversity. For example, in mammals, larger-bodied animals have more negative relationships with urbanization than smaller-bodied animals, but this relationship is intensified in cities that are characterized by warmer climates [24]. Furthermore, the level of taxonomic organization studied can impact the relationship between biodiversity and levels of urbanization. For example, a metanalysis of arthropod community responses to urbanization found an overall negative effect of increased urbanization on arthropod abundance and diversity, but the magnitude of this effect was much greater for specific orders of arthropods, with an increased negative effect in Coleoptera and Lepidoptera and a neutral response (i.e., mean effect size crossing zero) in Araneae [14]. These studies both demonstrate that the mechanisms underlying urban biodiversity are complex, and highlight the need for studies with wide geographic and taxonomic breadth to help us generalize patterns in ways that help cities improve their biodiversity [13,15,25,26].

Considering the projected increase in urban land cover within the next decade, the future of urban biodiversity will ultimately rely on the ability of global cities to attract and maintain populations of species that are largely considered urban intolerant. Well-planned cities can preserve and restore the habitat requirements of native species by facilitating heterogeneous landscapes, migratory stopover sites, and increased gene flow [27], among other initiatives. Likely as a result of urban planning efforts, some species' ranges have increased in urban areas over the past century [28–31]. Several studies have examined what factors increase and maintain urban biodiversity, for example, by evaluating the minimum number of native trees in urban residential yards needed to maintain diverse bird communities [32] or quantifying native species gained in planted rooftop gardens [33]. While these projects can help inform policy geared towards supporting and enhancing urban biodiversity, city managers still lack a comprehensive tool that can track spatio-temporal changes in urban biodiversity at the community level. As more native species are either threatened with extirpation or expand their ranges into urban environments, creating a tool that can track changes in urban diversity and community composition is more important than ever before.

In order to monitor diversity patterns and quantify the effects of varying levels of urbanization on different groups of native species, large amounts of data are needed across multiple

taxonomic groups and across broad areas of the urban environment. To correct for researcher biases that lead to datasets with limited geographic scope and taxonomic coverage, many studies have turned to large crowd-sourced datasets [15,34,35]. Such datasets are often referred to as 'unstructured' in that there is no required protocol for data collection, resulting in data that vary widely in their quality, organization, and information content [36]. One such platform, iNaturalist, has over 74 million observations for over 342,000 different species globally, 58% of which come from developed (i.e. urbanized) areas [37]. The size of the iNaturalist dataset gives it great potential for tracking and managing urban biodiversity. For example, Callaghan et al. [38] used community science data from metropolitan regions around Boston to quantify species- and community-level biodiversity responses to multiple urban gradients. These data were at a small-enough spatial unit to influence local policymaking. Large-scale public participatory datasets make urban biodiversity assessments at large spatial scales possible, even in cities, which tend to contain private lands that are largely excluded from structured biodiversity surveys [39].

Despite the abundance of data points from programs like iNaturalist, there are challenges associated with using these unstructured datasets to measure and manage urban biodiversity. For example, opportunistic sampling may lead to biases in data, as sampling effort is not equal across space, time, and taxonomic groups [40,41], potentially causing differences in user methodology to be misinterpreted as temporal or spatial changes in populations [42]. Analytical methods to best mitigate these inherent biases in unstructured data continue to be developed, including using both models and data processing to better account for unequal observations across space and time [43–45]. Particularly, these new methods use higher order taxa as indicators of survey effort which can inform negative occurrences and thus convert presence-only data to robust presence-absence format.

Here, we describe and implement an approach to spatially and temporally characterize urban tolerance of native species within the city of Los Angeles, California, USA using unstructured species occurrence data from iNaturalist. This approach was initially conceived to support the LA Biodiversity Index Baseline Report published by the Los Angeles Department of Sanitation and Environment [46] through the creation of an evaluative metric (Metric 1.2b; [47,48] that represents and monitors "Native Species Presence in Urban Areas." We refer to this index as a "Community Urban Tolerance Index" (CUTI), as it broadly aims to track how well native species that are often urban intolerant occur within Los Angeles by rating spatial units on the average urban association index or "UAI" (based on levels of urban tolerance) of their species assemblages. To assess this metric, we used iNaturalist data to estimate a species-level UAI for 967 species across six broad taxonomic focal groups that occur in Southern California. We then applied these indices to spatiotemporally thinned species occurrence data in order to calculate the CUTI for a spatial grid covering the city of Los Angeles. The CUTI represents the degree that the terrestrial animal community is composed of species that are either tolerant or intolerant to urbanization within a spatial unit. We then calculated a mean CUTI across all grid cells in Los Angeles, resulting in a single score for Metric 1.2b in the LA City Biodiversity Index. The methodology provided herein provides a framework for establishing repeated measures over time of urban tolerance within the city of Los Angeles and is applicable to other urban areas. Ultimately, these methods can help local managers and city officials across the region, state, or country understand and track the success (or failures) of local initiatives to support biodiversity and attract historically urban intolerant species to their cities. As urbanization poses a continued threat to biodiversity, particularly in biodiversity hotspots, the methods presented here will enable local governments to better manage and protect native biodiversity.

## Methods

### Study area

Our study was focused on Southern California, with an emphasis on the greater Los Angeles area, situated in the California Floristic Province, one of 36 biodiversity hotspots in the world [49,50]. The region is also one of five Mediterranean ecosystems in the world, which occur only on the western margins of landmasses between 30 and 40˚ latitude and which are typically characterized by cold and wet winters and warm and dry summers. Southern California has diverse topography, including the transverse mountain ranges to the north and east and the peninsular mountains ranges to the south, as well as diverse habitat types, including chaparral, coastal sage scrub, oak woodlands, coastal dunes and bluffs, riparian woodlands, and a variety of wetland habitats, which host over 2,200 species of vascular plants [51]. While the calculation of a formal CUTI was limited to areas within the city of Los Angeles, for the estimation of species-level UAI, our study area included all land within a 200 km buffer of the Los Angeles City boundary (approximate centroid: 34.031656 N, 118.241716 W), including the cities of Los Angeles, San Diego, Bakersfield, and Santa Barbara. We chose to focus on this broad geographic region because we were interested in creating a metric that could be measured repeatedly over time and would be robust to species that do not currently reside in our focal area of Los Angeles, but could colonize in the future. Furthermore, we treat urban tolerance here (as measured by the UAI) as a species-level trait, which is best estimated using occurrence data from a broader geographic region than just Los Angeles. As such, we aimed to include areas with a wide range of levels of urbanization, including multiple manifestations (i.e., cities) of urban land-use within the broader Southern California region.

### Urban intensity

In order to estimate urban associations, we first had to define a continuous spatial layer of the urban intensity of our study region. To do this, we used Principal Component Analysis (PCA), which decomposes multivariate datasets into major axes of variation, to create a single composite index of urban intensity from multiple sources. Following [38], this index included the Visible Infrared Imaging Radiometer Suite (VIIRS) nighttime lights data layer, but we added additional environmental variables related to urbanization as different taxa are likely to respond to different aspects of the urban environment. We initially tested a set of six data layers to depict urban intensity across our study area, but we removed several layers including pollution of fine particles smaller than 2.5 micrometers (PM2.5) [52], Average Traffic Volume [52], and Population Density [53] due to collinearity and coarser resolution. Thus, our PCA index represents a composite of three layers: (1) light pollution from VIIRS Version 4 DMSP-OLS Nighttime Lights Time Series (https://eogdata.mines.edu/products/dmsp/#v4_dmsp_download) [54], (2) "Percent Impervious Surfaces" from National Land Cover Database (NLCD) 2016 Percent Developed Imperviousness (CONUS) (https://www.mrlc.gov/data/nlcd-2016-percent-developed-imperviousness-conus) [55], and (3) "Noise Pollution" from National Park Service (NPS) Geospatial Sound Modeling 2013–2015 (https://irma.nps.gov/DataStore/Reference/Profile/2217356) [56]. All spatial layers were reprojected and resampled, as needed, to a 0.25 x 0.25 mile grid prior to combination. The first axis of the PCA explained 86.5% of the variance in all three layers (S1 Table), indicating as expected that the three layers are all indicative of the same general process (i.e., "urban intensity") yet individually add unique information. Because PCA axis 1 ("PC1") explained >70% of the variation, we retained it as our sole spatial index of urban intensity (Fig 1).

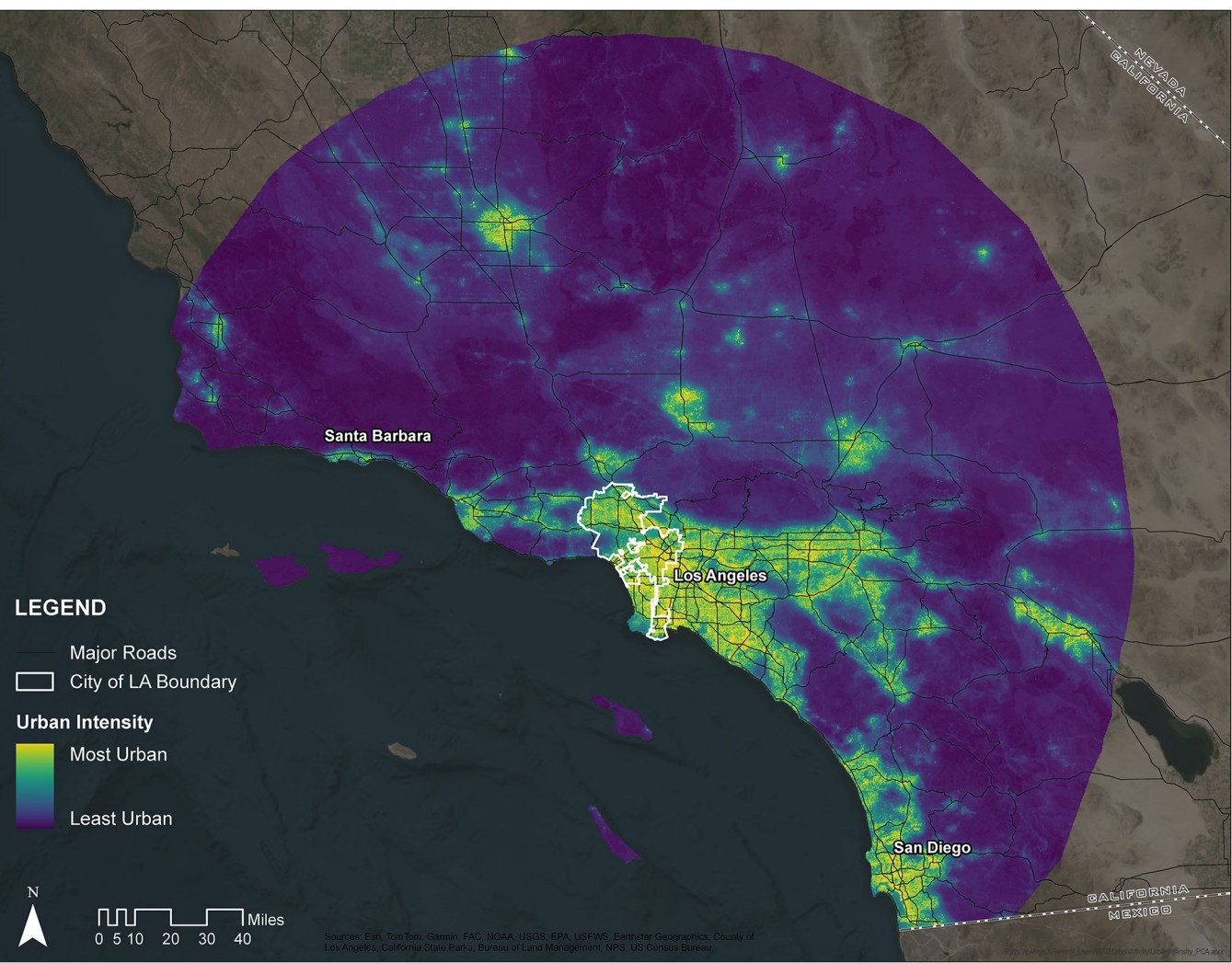

**Fig 1. Map of urban intensity measured across a broad study region in Southern California.** Our metric of urban intensity was determined as the first PCA axis of three different variables. Warmer colors indicate higher levels of urban intensity. Solid lines detail major roadways within California. Map created using data from [57–61].

## iNaturalist records and data quality filtering

We focused our analysis on selected target taxonomic groups, which were picked a priori with expert input to represent 12 taxonomic groups that are generally well-detected and well-surveyed by community scientists on the iNaturalist platform. The 12 groups include: snails and slugs (Order: Stylommatophora); spiders (Order: Araneae); dragonflies and damselflies (Order: Odonata); grasshoppers, locusts, and crickets(Order: Orthoptera); leafhoppers (Family: Cicadellidae); lady beetles (Family: Coccinellidae); hoverflies (Family: Syrphidae); bees and wasps (Family: Apidae and Vespidae); butterflies and moths (Families: Papilionidae, Pieridae, Lycaenidae, Nymphalidae, Sphingidae and Erebidae); herpetofauna (Classes: Amphibia and Reptilia); mammals (Class: Mammalia); and birds (Class: Aves). We queried the iNaturalist API for occurrence data on 13 January 2022 using the 'rinat' package version 0.1.8 [62]. We downloaded all iNaturalist records between 2011–2021 for the higher order taxa groups defined above, only limiting records to "research quality" georeferenced occurrences (i.e.,

those with a consensus taxonomic ID and location coordinates) bound within our broader study region (i.e., 200 km radius around Los Angeles). After downloading, we further filtered our data to remove species that had more than 60% of all observations records marked as "Geoprivacy = obscured" (a situation where iNaturalist provides spatial coordinates of sightings, but these coordinates are randomly offset by up to 22 km from the true location). Although we additionally filtered out all obscured spatial records for all species, we wholly excluded species meeting this arbitrary threshold as we believed that such a widespread degree of geoprivacy indicated species for which remaining iNaturalist data would not likely represent the species' true distribution within the study area. Finally, we reclassified all records identified to the subspecies level to the species level following other similar work using related datasets [63,64].

## Expert review

After downloading iNaturalist records and applying our initial hard filters for data quality, we engaged the LASAN Biodiversity Expert Council, a regional group of scientists and taxonomic specialists who advise the annual biodiversity report, to further assist in data curation and QA/QC. Five experts in areas of specific taxonomic focus for the species included in this study (i.e., arthropods, mollusks, mammals, birds, and herpetofauna) were asked to evaluate occurrence data for the specific higher order taxonomic grouping they specialized in by using the following questions: 1) Is this species native to the study area?; 2) Is the species terrestrial?; 3) Is the natural history of this species so different from others in its taxonomic grouping that records for this particular species should not be used as indicators of search effort for other similar species?; and 4) Is there any other reason why we should exclude this species from this study? The third question refers to the issue that iNaturalist data are presence-only and do not, on their own, provide information on absence or non-detection. Increasingly, however, ecologists are using multi-taxa presence-only surveys to bin species into 'detection groups', whereby an observation of one species at a location provides inference on the non-detection of other species [44,65]. This assumption of substitutability is justified as natural history observers are often searching broadly within taxonomic groups; for example, a birdwatcher's positive record of one bird species says more about the non-detection of another bird species than it does about the non-detection of a butterfly. In the context of the present study, we did not require species' occurrences to be perfect indices of non-detection for other species, but simply sought taxonomic groups where the presence of one species in that group would serve as a broad index of survey effort for all species in that group. Thus, we sought via expert review to exclude taxa that differed so much from the rest of their grouping (e.g., diurnal versus nocturnal; or identifiable via photography versus identifiable only via microscope) that they should not be treated as survey effort proxies. For the fourth question, some common reasons for excluding species based on expert review included species with extremely limited distributions that would otherwise be uninformative to urban tolerance (e.g., a species of plethodontid salamander limited to a single remaining population on Mt. Baldy, Los Angeles, USA), or misidentifications based on recent taxonomic splits.

Following data review by taxonomic specialists, we curated their responses to make sure that experts interpreted these questions similarly. We filtered observations based on these responses to exclude non-native species, species unlikely to be detected by typical observers, non-terrestrial (i.e. marine or freshwater) species, and species according to additional criteria as determined by the taxonomic group specialists.

## Controlling for differences in sampling effort

We took a number of steps to control for differences in sampling evenness and effort in iNaturalist data. First, to address the inherent biases associated with sampling evenness, we performed a broad spatiotemporal thin [66,67]. We thinned species-specific data to one observation per year within each 0.25 x 0.25 mile grid cell. This produced a database where every species is recorded as present or not detected in each grid cell and for each year between 2011–2021 (i.e. the number of yearly detections out of 11 years). Second, to address the additional bias associated with varied sampling effort, we defined site-specific sampling effort for each of the twelve focal taxonomic groups. We did this because observers may not equally record observations for all taxonomic groups (e.g., an observer may not observe spiders while recording birds). Taxonomic group-specific effort grids were calculated based on the observation effort per year for the corresponding focal taxonomic groups, summing within grid cells the number of years with at least one observation in a year of a species from within a taxonomic group. Thus, taxonomic group-specific effort layers indicate the number of years (0–11) in which observers obtained at least one record of a target group, which serves as the maximum potential number of thinned presences for any given species of that group. As such, the thinned species presence layer and the matching effort layer represent a spatially-varying binomial response, where the number of binomial trials is the effort in a grid cell and the number of binomial successes is the thinned number of species' presences.

## Measuring species-level relationship to urban intensity layer in study area

Species-level indices of urban association were calculated based on thinned occurrence data and taxon-matched effort data across the entire Southern California study region. We calculated a UAI for each species that had at least 25 thinned annual occurrences across our study region. Specifically, using the 'stats' package [68], for each species we modeled the number of thinned occurrences, given effort per cell, as a binomial process that varied as a function of a single covariate: the urban intensity (PC1) of the grid cell. This model allowed us to estimate the number of thinned species occurrences as a binomial variable, where the number of successes (i.e., 'occurrences') was capped by the number of years with non-zero effort for the taxonomic group in each cell. In this way, our model accounted for taxon-specific sampling effort over time in each grid cell. The resulting logit-linear slope of the trend line, which indicates the relationship between species' occurrence and urban intensity, was stored as the UAI for a given species. Positive slopes indicate urban tolerance, negative slopes suggest urban intolerance, and slopes of zero indicate no relationship of occurrence to urban intensity.

## Calculating a Community Urban Tolerance Index (CUTI)

After calculating a UAI for each species, we quantified a CUTI for each grid cell in Los Angeles by taxonomic group, as well as a composite score for the entire city (i.e., metric 1.2b for the city of Los Angeles). To calculate a taxonomic group-specific CUTI for each grid cell in Los Angeles, we matched species' UAIs to species occurrences in individual grid cells and calculated a raw CUTI score per cell by taking the mean UAI score of all species within any given cell, weighted by the thinned temporal occurrence of each species (i.e., a value of 1–11 for the number of years that the species occurred in that cell). This resulted in 12 grids (one for each taxonomic group) with a group-specific CUTI score for every cell in the city in which the group was detected. To interpret these results at a broader taxonomic scale, we also calculated the mean CUTI across all pixels for each of the 12 taxonomic groups. Finally, to calculate a composite CUTI across all taxonomic groups, we averaged the 12 taxonomic group grids and city-wide scores. In all cases, the raw CUTI scores were binned into a 5-point scale as follows:

-Infinity to -0.5 = 5; -0.5 to -0.25 = 4; -0.25 to 0 = 3; 0 to 0.25 = 2; 0.25 to 0.5 = 1; and 0.5 to Infinity = 0. On this scale, a cell with a CUTI index of 4 or 5 suggests that species in aggregate are more natural-area associated, while a cell with an index of 0 or 1 suggests that species are more urban tolerant. To test for an association between UIA and urban intensity (with the hypothesis that areas of higher urban intensity have lower CUTI scores), we used an ANOVA and Tukey HSD test in the 'stats' package [69] with urban intensity as the response variable and categorical binned CUTI scores for each grid cell as an independent variable.

## Results

Our iNaturalist query yielded a total of 958,624 observations from 127,553 observers (Table 1; [64]). After filtering these observations, we retained 567,996 observations from 71,120 observers. Our filtered query included a total of 967 unique native species found within the study area, of which 563 occurred at least once within the city of Los Angeles. We were able to calculate UAI for 510 species in our dataset, of which 408 occurred at least once within the city of Los Angeles. The species assessed were on average negatively associated with our measure of urbanization, although there was variation across species (cross-species mean = -0.21, range = -2.93 to 0.62; Figs 1 and 2, & S2 Table). UAI varied between the 12 taxonomic groupings (Table 2), with snails and slugs having the highest (i.e. more urban tolerant) score (group mean = 0.24, range = -0.096 to 0.62), and butterflies and moths having the lowest (i.e. more urban intolerant) UAI (group mean = -0.40, range = -2.93 to 0.46). The most urban associated species in our study was the slipper snail (*Cochlicopa lubrica*) (UAI = 0.62), and the least urban associated species was the greenish blue butterfly (*Icaricia saepiolus*) (UAI = -2.93).

**Table 1. Counts of observations and observers between unfiltered dataset downloaded from iNat API using specific identifiers for higher order groupings (i.e., iNat Taxa ID) and iNaturalist data subject to exclusion by expert review (see methods for criteria) and hard filters resulting in a filtered dataset of native species.**

| Taxa Information | | | Unfiltered | | Filtered | | |
|---|---|---|---|---|---|---|---|
| **Higher Order Grouping** | **Taxon** | **iNat Taxa ID** | **# Observations** | **# Observers** | **# Observations** | **# Observers** | **# Species** |
| Snails and Slugs | Stylommatophora | 47485 | 22,539 | 5,507 | 919 | 213 | 5 |
| Spiders | Araneae | 47118 | 28,148 | 8,897 | 16,029 | 5,778 | 144 |
| Dragonflies and Damselflies | Odonata | 47792 | 16,391 | 3,431 | 16,366 | 3,426 | 73 |
| Grasshoppers, Locusts, and Crickets | Orthoptera | 47651 | 17,267 | 5,670 | 13,816 | 4,483 | 140 |
| Leafhoppers | Cicadellidae | 53237 | 2,585 | 849 | 796 | 236 | 41 |
| Lady Beetles | Coccinellidae | 48486 | 18,030 | 5,090 | 9,246 | 2,677 | 37 |
| Hoverflies | Syrphidae | 49995 | 10,654 | 2,200 | 8,212 | 1,832 | 40 |
| Bees and Wasps | Apidae | 47221 | 34,748 | 11,105 | 11,499 | 3,166 | 68 |
| | Vespidae | 52747 | 5,113 | 1,826 | | | |
| Butterflies and Moths | Erebidae | 121850 | 3,939 | 1,714 | 28,480 | 6,973 | 68 |
| | Lycaenidae | 47923 | 15,199 | 2,479 | | | |
| | Nymphalidae | 47922 | 38,786 | 10,023 | | | |
| | Papilionidae | 47223 | 8,514 | 3,963 | | | |
| | Pieridae | 48508 | 9,528 | 2,504 | | | |
| | Sphingidae | 47213 | 6,988 | 3,967 | | | |
| Herpetofauna | Amphibia | 20978 | 19,892 | 5,229 | 72,312 | 13,992 | 42 |
| | Reptilia | 26036 | 70,637 | 13,601 | | | |
| Mammals | Mammalia | 40151 | 56,914 | 11,763 | 34,107 | 7,514 | 49 |
| Birds | Aves | 3 | 572,752 | 27,735 | 356,214 | 20,830 | 260 |
| | | **TOTAL** | 958,624 | 127,553 | 567,996 | 71,120 | 967 |

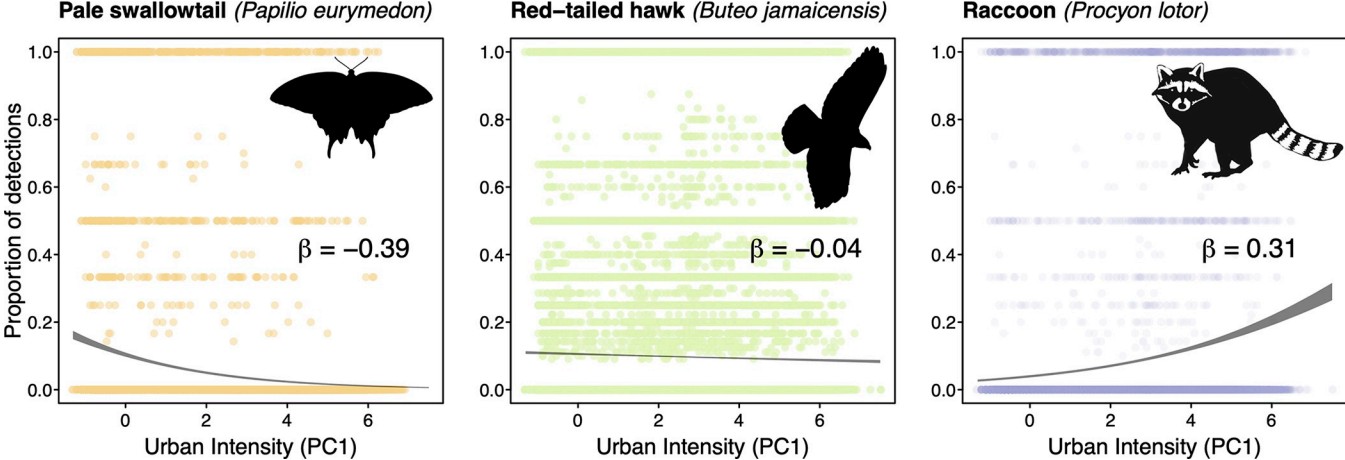

**Fig 2. Species vary widely in their response of occurrence to urbanization.** Scatterplots of three different taxa showing urban intolerance (left), urban ambivalence (center), and urban tolerance (right). Scatterplots show the proportion of detections (out of a maximum of 11 years) for each species across each grid cell in the broader Southern California study region (Fig 1). Trend lines show the 95% confidence interval surrounding a binomial regression of detection frequency as a function of urban intensity. Species' UAI scores ($\beta$) are the logit-linear slope of the trend line.

We assessed 8,348 0.25 x 0.25 mile grid cells across the city of Los Angeles for their CUTI, weighted by the temporally thinned occurrence of each species. A total of 2,010 grid cells did not have records for any target taxa after filtering and were not included in our calculation of summary scores. Averaging across all higher order taxonomic groupings, the city had an average CUTI of 2.01 (raw, unbinned score = 0.129; Fig 3). Average CUTI varied between higher order taxonomic groupings (Table 3), with snails and orthoptera demonstrating the highest average CUTI (snails: 1.40 binned, 0.30 raw; orthoptera: 1.54 binned, 0.17 raw), and odonates and mammals showing the lowest average CUTI (odonates: 2.25 binned, 0.06 raw; mammals: 2.16 binned, 0.07 raw) (Fig 4). There was a significant relationship between urbanization values and the cross-taxa average CUTI of those cells, with areas of higher urban intensity holding taxa that, on the whole, were more urban tolerant (i.e, have higher UAI values) (ANOVA, p < 0.001; S1 Fig). This general relationship held true for every individual taxonomic group (ANOVA, p < 0.001), except for snails (ANOVA, p = 0.99).

**Table 2. Average urban association index (UAI) scores for each of the 12 higher order taxonomic groupings.** Species-level UAI scores ranged from -2.9 to 0.62, with more negative numbers indicating more urban intolerant species and more positive scores indicating more urban tolerant species. Range of values in parentheses.

| Higher Order Grouping | Average UAI Score (range of all species) |
|---|---|
| Snails and Slugs | 0.24 (-0.10 to 0.62) |
| Spiders | -0.14 (-0.90 to 0.62) |
| Dragonflies and Damselflies | -0.20 (-1.32 to 0.37) |
| Grasshoppers, Locusts, and Crickets | -0.37 (-2.20 to 0.36) |
| Leafhoppers | -0.08 (-0.51 to 0.34) |
| Lady Beetles | -0.08 (-0.63 to 0.37) |
| Hoverflies | -0.11 (-0.54 to 0.27) |
| Bees and Wasps | -0.16 (-0.83 to 0.43) |
| Butterflies and Moths | -0.40 (-2.93 to 0.46) |
| Herpetofauna | -0.34 (-1.32 to 0.35) |
| Mammals | -0.39 (-2.20 to 0.31) |
| Birds | -0.15 (-1.79 to 0.47) |

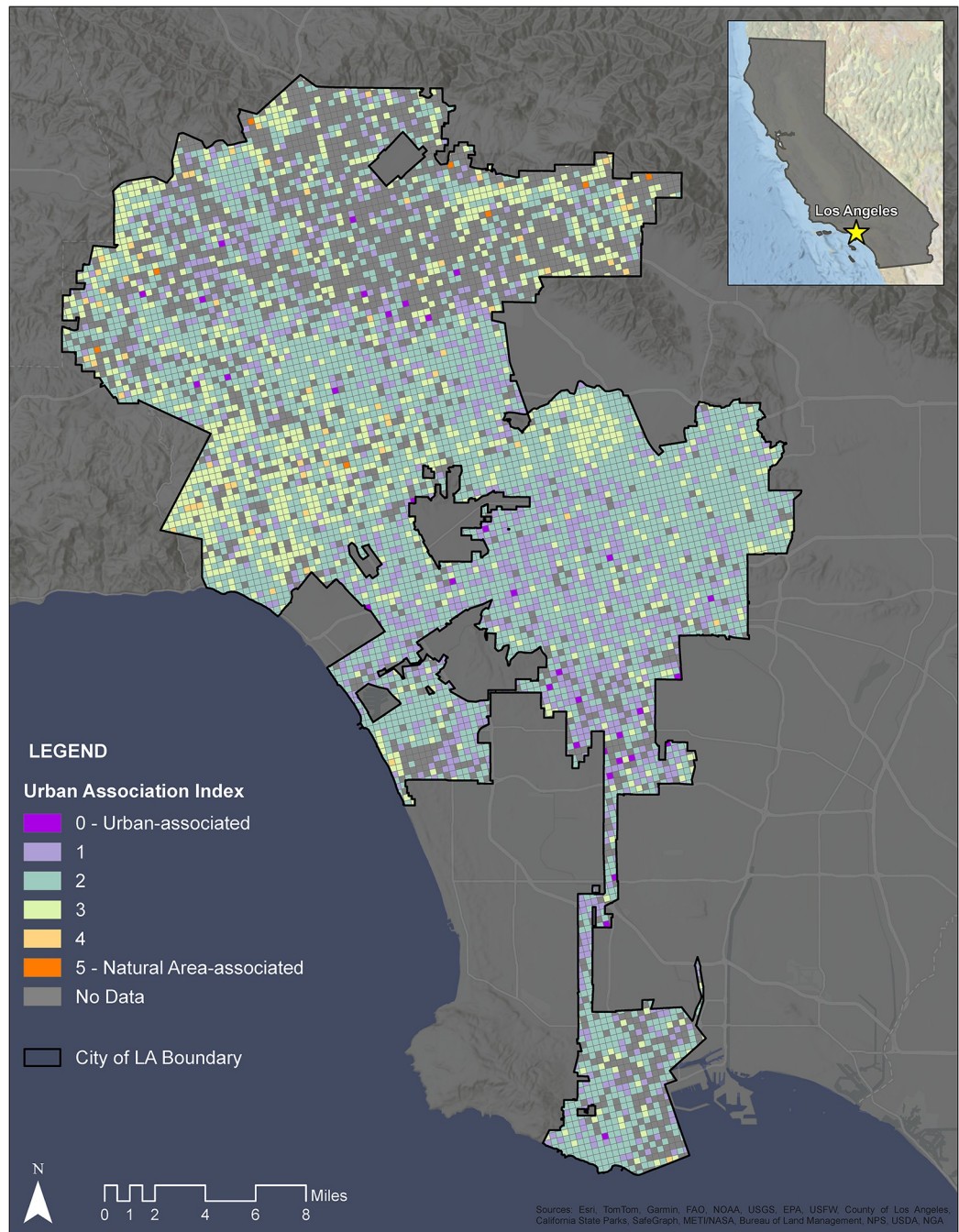

**Fig 3. Map of the City of Los Angeles with overlaid mean composite urban association (UAI) scores.** Warmer colors indicate more natural area tolerant species, whereas cooler colors indicate more urban tolerant species. Areas within Los Angeles city boundaries with insufficient data to calculate the score are colored gray. Map uses a 0.25 x 0.25 mile scale.

## Discussion

Using over 567,000 publicly available community science records from iNaturalist, we present the first comprehensive species-level evaluation of urban tolerance for Southern California taxa. For the City of Los Angeles, we found that on average, native species within the city were negatively associated with our measure of urban intensity (i.e. light pollution, impervious

**Table 3. Community urban tolerance index (CUTI) scores for each of the 12 higher order taxonomic groupings.**
Raw CUTI values are the average of weighted average of species-level UAI scores, while binned CUTI values rescale to a 5-point index, where a CUTI index of 4–5 suggests that species in aggregate are more natural-area associated, while an index of 0–1 suggests that species are more urban tolerant.

| Taxa | Raw CUTI | Binned CUTI |
|---|---|---|
| Snails and Slugs | 0.30 | 1.40 |
| Spiders | 0.22 | 1.63 |
| Dragonflies and Damselflies | 0.06 | 2.25 |
| Grasshoppers, Locusts, and Crickets | 0.17 | 1.54 |
| Leafhoppers | 0.10 | 2.15 |
| Lady Beetles | 0.12 | 1.98 |
| Hoverflies | 0.10 | 1.95 |
| Bees and Wasps | 0.20 | 1.81 |
| Butterflies and Moths | 0.13 | 1.96 |
| Herpetofauna | 0.14 | 1.99 |
| Mammals | 0.07 | 2.16 |
| Birds | 0.10 | 2.15 |

surfaces, and noise pollution), and that this pattern was conserved across the higher order taxonomic groupings with the exception of snails and slugs, which were positively associated with urban intensity. We then averaged this species-level response to our measure of urban intensity on a ¼ mile grid across the City of Los Angeles to understand the geographic distribution of urban intolerant species. Ultimately, this metric of native species urban tolerance (CUTI) can be reassessed regularly as a means of evaluating change in urban tolerance over time and following specific biodiversity improvement measures in the city.

## Extrinsic and intrinsic factors leading to differences in UAI

The most urban tolerant species in our study was the slipper snail (*C. lubrica*). This species is known to be widespread and euryhygric (i.e. able to withstand a broad range of moisture conditions), and it is possible that urbanized areas such as Los Angeles provide year-round access to a variety of moisture regimes through the addition of ornamental landscapes and lawns [70–72]. Other studies have found high native snail abundance in areas of high urban intensity in Tennessee [73]. Given that there were only five species of snails and slugs that remained in our dataset post-filtering, this pattern could largely be due to a small sample size of species, and perhaps more purposeful sampling is required to truly ascertain the affinity of this entire taxonomic group for urban environments.

The most urban intolerant species in our study was the greenish blue butterfly (*Icaricia saepiolus*). Compared to other North American butterfly families, Lycaenidae is overrepresented in terms of number of species proposed for listing [74]. This is largely due to host plant specificity of Lycaenids, primarily for plants in the genera *Lupinus* and *Eriogonum*, and the fact that these plants are adapted to disturbance regimes that are infrequent in the urban context [74–76]. Conservation of many of the special status butterfly species including several Lycaenids therefore relies on maintaining and expanding critical segments of habitat that contain host plant species within urban settings and maintaining habitat fragments of varying sizes through deliberate disturbance as a management tool [76]. For example, the Palos Verdes blue butterfly (*Glaucopsyche lygdamus palosverdesensis*) is an endangered subspecies of Lycaenid butterfly in Los Angeles County, and findings from the US Fish and Wildlife Service demonstrate that the species appears to be establishing in reintroduction sites due primarily to efforts

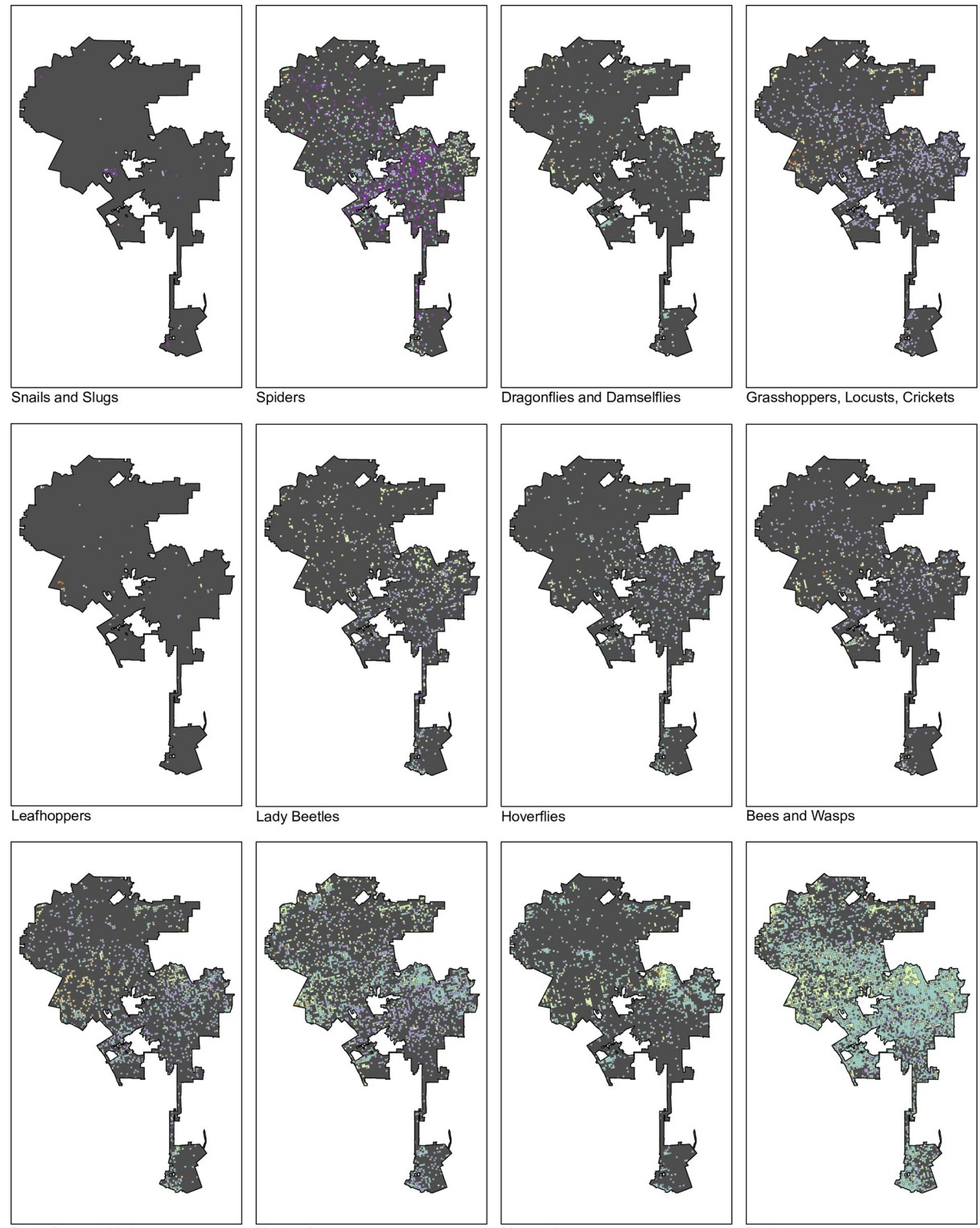

**Fig 4. Map of the urban association scores separated by higher order taxonomic groupings.** Levels correspond to Fig 3, where warmer colors indicate more urban intolerant species, cooler colors indicate more urban tolerant species, and gray indicates cells with insufficient data to calculate a score.

to remediate historical habitat through mechanical disturbance, non-native plant removal, and planting of successional host plants [77]. Based on the findings from this study, the City of Los Angeles could use existing butterfly observations to identify target areas for conservation for the Palos Verdes blue butterfly and other imperiled butterfly species. In areas where observations overlap with property owned and managed by the City of Los Angeles, the city can focus restoration efforts on increasing host plant abundance through direct plantings of early successional plants and targeted mechanical disturbance to create conditions necessary for early successional habitat needed by the butterflies. Following restoration activities like these, city managers can evaluate success through a reevaluation of this metric on an annual basis.

All other higher order taxonomic groups had higher presence in less urbanized regions within the city. Previous research in Los Angeles, research has demonstrated that the presence of many taxonomic groups is negatively affected by increased levels of urbanization. For example, coyotes and bobcats in mixed urban/natural areas have home ranges that primarily utilize natural areas [78], and some regionally common amphibian species are markedly absent from streams within urbanized areas of the city [79], but within taxonomic groupings there was high levels of variation in individual species-level responses. Many factors contribute to these varied responses of species to different levels of urban intensity, including species-specific functional traits. Specifically, in mammals and birds, functional traits help explain urban tolerance of species, including body size, dietary breadth, clutch size, and nesting strategy, among other predictors in birds [64] and litter size in mammals [80]. While our study does not seek to evaluate all individual species-level traits and how they relate to values of the UAI, some previously noted relationships between species' functional traits and responses to urbanization are recapitulated in our findings. For example, Cooper et al. [81] found that urban raptors in Los Angeles including Cooper's hawk (*Accipiter cooperii*) and red-shouldered hawk (*Buteo lineatus*) responded to increases in urban cover within their home ranges by increasing nesting in urban sites, while other species of urban raptors such as American kestrels (*Falco sparverius*) were more likely to nest in less-urban areas within their home ranges. They concluded that nesting strategy may play a role in the response of urban raptors to levels of urbanization, as American kestrels are cavity nesters and dead trees are likely to be removed by homeowners [81]. Notably, these previously-reported responses to urbanization for Cooper's hawks, red-shouldered hawks, and American kestrel are confirmed in our estimated values of UAI (0.11, 0.19, and -0.03 respectively). Future studies could further explore the relationships between species-specific functional traits and UAI for the 510 native species studied here using existing databases of functional traits such as AVONET [82], AnimalTraits [83], or COMBINE [84].

In addition to intrinsic characteristics of species, many extrinsic factors may contribute to species-level responses to varying levels of urban intensity. For example, research on urban insect populations in Los Angeles found that diurnal temperature range had a consistent negative effect on occurrence of all arthropod species studied [85]. We found a negative association with urbanization across all native arthropods, although we also observed high levels of variation within higher order taxonomic groupings. It is possible that spatial variation in water resources across the city can partly explain this within-group variation, as has been demonstrated for desiccation-sensitive invasive insect species in Los Angeles [86]. The heterogeneous presence of water on the landscape also strongly influences the vegetation community present, which in turn strongly influences arthropod community composition. Additional research could build off our findings to investigate the relationship between our measures of UAI and both spatial variation in abiotic gradients and functional traits of individual species, such as desiccation tolerances of insects. The large amount of variation within groups can perhaps also be explained by different species-level responses to the anthropogenic stressors that come with urbanization, namely light pollution and noise pollution. Because spatial datasets of artificial

light at night (ALAN) and anthropogenic noise were used in the creation of our urban intensity layer for this study, we know that these measures of anthropogenic stressors explain much of the variation in our dataset when considered in aggregate (S1 Table). This is unsurprising given the broad impacts of ALAN [87] and anthropogenic noise [88] for many different groups of organisms, including many of the higher-order taxonomic groupings studied here, such as birds (light: [89]; noise:[90]), mammals (light: [87]; noise: [91]), and invertebrates (light: [92]; noise: [93]). These stressors can lead to a variety of responses that are specific to individual species or communities, including some positive responses (e.g., some bats species have increased foraging opportunities as a results of ALAN; [94]). Finally, large within-group variation in UAI values may also be related to the variable, and relatively artificial, taxonomic levels that we used to aggregate data; future studies with more fine resolution groupings (e.g., at the genus level) may reveal more taxonomically conserved and biologically relevant relationships.

## Limitations of and future directions for using crowd-sourced data

While this study includes over 500 native species observed within the study area, approximately 59% of the total community science records were excluded from the analyses because they were considered undetectable to the general public (e.g. small insects, nocturnal mammals, etc.), the data were not at the "research grade" level, or the records for a given species were geoobscured due to species status or user preference. Other studies have noted similar data quality issues and biases in iNaturalist records [95–97]. In order to circumvent data quality issues, this study relied on expert review to identify higher order taxonomic groupings that could be reliably identified by the general public, but in doing so, may have increased ascertainment bias and decreased the overall scope of the data. In an effort to reduce the amount of data lost, future assessments of this metric may benefit from developing a relationship with iNaturalist in order to obtain user-obscured data en masse, which is not currently possible without requesting thousands of individual records from each iNaturalist user. This study may also be limited by bias within higher order taxonomic groups for species that are common or more easily observed, as has been reported previously [98,99]). While we were unable to confirm whether any given species is present within our dataset more or less often than "true" occurrence, due to a relationship with their abundance on the landscape or other factors, we believe that this bias should act randomly across the study area and therefore not impact the overall interpretability of our findings. Additional work could greatly benefit the field by investigating the potential over- or underrepresentation of common species within community science datasets.

While we present several limitations to the available crowd-sourced species presence dataset within our sampling area, these data limitations also provide targets for local environmental managers to improve these datasets and therefore biodiversity monitoring in their regions. Based solely on the number of assessed grid cells across the City of Los Angeles in this study, it is clear that there needs to be substantial effort placed on bolstering community science projects that focus on underrepresented taxonomic groups (e.g. snails and slugs and leafhoppers). Findings from other community science projects indicate that local city residents are underrepresented contributors to community science datasets [100], yet for community science-based biodiversity monitoring to be successful, it must be built from a bottom-up approach that includes both participatory and contributory opportunities for the communities where biodiversity monitoring is to take place [101]. While some efforts in Los Angeles to involve residents in taxonomically focused community science projects have led to increased knowledge of urban biodiversity for these taxa groups (e.g. the BioSCAN project, see https://nhm.org/community-science-nhm/bioscan; [102]), these projects are limited in geographic scope, are

often short-term assessments, and the primary role of residents is that of data collection, which may not lead to sustained participation by community members in the future [103]. Moreover, efforts to educate the public on specific indicator species is underway in Los Angeles [104], and these efforts could be directed at underrepresented taxonomic groups highlighted in this study. It has been demonstrated that community science datasets can match if not surpass traditional biodiversity assessment methods in data quantity, and do so in a fraction of the time [105,106]. Therefore, developing long-term and mutually beneficial partnerships with local communities to assess urban biodiversity should be a primary focus of city managers who plan to use large unstructured community science datasets to measure the efficacy of city-wide biodiversity measures.

## Conclusions

Herein we present a broad taxonomic assessment of the urban tolerance of native animal species for the City of Los Angeles. We found that there are clear differences in species level responses to our measures of urban intensity and that native species within Southern California are largely urban intolerant. This is even more true within the City of Los Angeles. This study provides a baseline assessment of the degree of presence of urban intolerant species within the City of Los Angeles in a ten year study period. Repeated assessment of this metric will allow stakeholders such as the City of Los Angeles to monitor success of its stated goal of no-net biodiversity loss by 2035. An important metric within the City's Biodiversity Index is the ability of this urban system to attract and maintain healthy populations of urban intolerant species.

## Supporting information

**S1 Fig. Regression plot of urban intensity (first PCA axis) and CUTI scores for all 510 species.**
(TIF)

**S1 Table. Table detailing the loadings and variance explained for the composite urban intensity layer separated by contributing layers and PC axes.**
(DOCX)

**S2 Table. Table of all species considered in this paper and their associated urban affinity scores.**
(CSV)

## Acknowledgments

We would like to acknowledge Dan Cooper, Miguel Ordenana, and Jessica West for providing us access to iNaturalist project data. The study authors would like to specifically thank Gary Bucciarelli, Rachel Chock, and Jann Vendetti for their expert review of iNaturalist data of herpetofauna, mammals, and slugs and snails, respectively. The study authors would also like to acknowledge the LA City Biodiversity Expert Council, and particularly the members who attended the workshop for Metric 1.2b. Finally, we thank the iNaturalist observers and identifiers for their contributions that made this research possible.

## Author Contributions

**Conceptualization:** Joseph N. Curti, Michelle Barton, Rhay G. Flores, Maren Lechner, Alison Lipman, Albert Y. Park, Morgan W. Tingley.

**Data curation:** Joseph N. Curti, Graham A. Montgomery, Morgan W. Tingley.

**Formal analysis:** Joseph N. Curti, Morgan W. Tingley.

**Methodology:** Michelle Barton, Morgan W. Tingley.

**Project administration:** Joseph N. Curti, Michelle Barton, Alison Lipman, Morgan W. Tingley.

**Supervision:** Joseph N. Curti, Michelle Barton, Alison Lipman, Morgan W. Tingley.

**Visualization:** Kirstin Rochel, Morgan W. Tingley.

**Writing – original draft:** Joseph N. Curti.

**Writing – review & editing:** Joseph N. Curti, Michelle Barton, Rhay G. Flores, Maren Lechner, Alison Lipman, Graham A. Montgomery, Albert Y. Park, Morgan W. Tingley.

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
