## [Decision Letter · Decision Letter 0]

26 Jan 2024

PONE-D-23-39010Using Unstructured Crowd-sourced Data to Evaluate Urban Tolerance of Terrestrial Native Species within a California Mega-CityPLOS ONE

Dear Dr. Curti,

Thank you for submitting your manuscript to PLOS ONE. After careful consideration, we feel that it has merit but does not fully meet PLOS ONE’s publication criteria as it currently stands. Therefore, we invite you to submit a revised version of the manuscript that addresses the points raised during the review process.

We have received two external revisions. I also revised the manuscript.Reviewer 1 suggested Minor Revision, and provided several compliments on your study. He/she provided a set of suggestions and corrections to improve the quality of your study.

Reviewer 2 suggested Minor Revision, and provided a range of suggestions/corrections trying to improve your manuscript.

 I ask you to try to follow the suggestions/corrections that you consider appropriate, and justify when disagree with them. Please find their reviews and mine below.

We look forward to receiving your revised manuscript.

Kind regards,

Dárius Pukenis Tubelis, Ph.D.

Academic Editor

PLOS ONE

4. We note that Figures 1,3 and 4 in your submission contain [map/satellite] images which may be copyrighted. All PLOS content is published under the Creative Commons Attribution License (CC BY 4.0), which means that the manuscript, images, and Supporting Information files will be freely available online, and any third party is permitted to access, download, copy, distribute, and use these materials in any way, even commercially, with proper attribution. For these reasons, we cannot publish previously copyrighted maps or satellite images created using proprietary data, such as Google software (Google Maps, Street View, and Earth). For more information, see our copyright guidelines: http://journals.plos.org/plosone/s/licenses-and-copyright.

a. You may seek permission from the original copyright holder of Figures 1,3 and 4 to publish the content specifically under the CC BY 4.0 license. 

5. Please upload a copy of Supporting Information (Supplementary Table 1 and 2 and Supplementary Figure 1) which you refer to in your text on page 21.

Additional Editor Comments:

Review by the Editor (Dárius Tubelis):

The manuscript has several problems regarding formatting to PLOS ONE. Please review the Submission Guidelines (https://journals.plos.org/plosone/s/submission-guidelines), and follow the instructions regarding this aspect in every section (below).

General:

Manuscript text should be double-spaced.

You have to use a numerical system for citations. See recent PLOS ONE papers. Examples:

[2] note the square brackets. For a single ref.

[4, 5]…….[3, 6] note the space after the comma. For two refs.

[4 – 6]…..use a long dash, not hyphen, without space. For 3 or more refs.

Please find some comments, corrections and suggestions for specific sections:

Abstract

Well done. Use “Abstract” (only the first letter in capital).

Introduction

Use “Introduction”. Only the first in capital. The same for other top section titles.

Very good content and length. Objectives were well explained.

Methods

Use “Methods”

And “Study area” in bold, not italics.

Study Area

It would be good to provide some brief information, in a few sentences, about the environment of this region, with references (e.g. climate, terrain, original vegetation, landscapes….). You will have an international readership.

Line 164. Subtitle. In bold, not italics.

Line 184. S1 Table. I could not find its link. Have you added this to the submission ?

Line 187. Use “Fig 1” (without the dot).

Line 196. In bold, not italics. The same for other subtitles.

Lines 202-205. Should these family names be in italics ? I think not. Please check this. Text and tables.

Results.

There is no need for bold when citing figures and tables. Only in captions and titles.

Line 326. Use “Fig. 1, Fig 2, S2 Table). Have you included this table in the manuscript ? There is no link.

Table 1. You have to provide one or more words that comprise the top names of columns 1-3. Such as “Taxa information” ?

Line 363. Have you included S1 Fig in the manuscript ? Use “S1 Fig”.

Linne 365. Something was cut. Please complete.

Discussion

You could consider adding subtitles along the discussion to ease the reading.

Line 427. Maybe start a new paragraph.

Your discussion miss an ecological section….try to add 1-2 paragraphs to discuss factors that could influence the positive, neutral and negative responses of different animals to urbanization. Excessive noise/light, lack of resources, presence of gardens, urban parks. Given attention to the three components of urbanization.

Supplementary material

Apparently, your submission does not have it.

You have to add them when submitting the corrected version.

Here in the end of the ms, you have to add the titles of S1 tables and captions of S1 figures.

References.

Please check again Submission Guidelines and recent PLOS ONE papers for correct formatting. Some major problems:

This alphabetical order is not acceptable. You have to list the references according to the order in which they appear along the text.

You have to add DOI for all references that bring this information. Format: https://doi.org/10.....

For refs with journals, It should be: Journal. Year; volume(issue):pages. Months are not necessary.

Titles of journals should be abbreviated.

For several refs, you placed the first letter of words in capital; it should not occur, except for proper names, etc…

Check also books, chapters and webpages, etc…..

Reviewers' comments:

Reviewer's Responses to Questions

**Comments to the Author**

1. Is the manuscript technically sound, and do the data support the conclusions?

Reviewer #1: Yes

Reviewer #2: Yes

2. Has the statistical analysis been performed appropriately and rigorously? 

Reviewer #1: Yes

Reviewer #2: Yes

3. Have the authors made all data underlying the findings in their manuscript fully available?

Reviewer #1: Yes

Reviewer #2: Yes

4. Is the manuscript presented in an intelligible fashion and written in standard English?

Reviewer #1: Yes

Reviewer #2: Yes

5. Review Comments to the Author

Reviewer #1: Summary: This manuscript details a study evaluating the urban tolerance of native animal species in Los Angeles, California, using occurrence data from iNaturalist. Using a composite urbanization rank, 510 native animal species were evaluated for their urbanization associations and animal communities were evaluated for their urban tolerance across LA. The authors found that urbanization responses were taxa-specific, with snails and slugs were most positively associated with urbanization and butterflies and moths were least. This manuscript is well written, comprehensive in its scope, and has a strong analytical design, especially regarding the use of crowd-sourced data. Few studies take such a taxa-inclusive approach to describing patterns in community composition and occurrence across urbanization gradients, which will make this paper stand out in the literature. I commend the authors on such a strong submission. Below are a few minor points to address for clarity.

Title: May want to indicate that terrestrial native animal species were the focus of this paper.

Line 36: The statement “e.g., mammals are not the same as birds” seems unnecessary to specify here.

Lines 60–76: Regarding taxa-specific responses to urbanization, I would suggest reading Hahs et al. 2023 “Urbanisation generates multiple trait syndromes for terrestrial animal taxa worldwide” in Nature Communications. Many of the same taxonomic groups were sampled in this manuscript and Hahs et al. regarding their functional traits and urbanization associations. Right now, this paragraph and the next heavily focus on birds and their associations with plants in cities. While birds have historically been one of the major animal taxonomic groups studied in the urban biodiversity literature, there were 11 more taxonomic groups surveyed in this paper that are worth mentioning here.

Line 62: Rega-Brodsky et al. 2022 did not report an association between urbanization and loss of plant diversity, only that the bulk of urban biodiversity literature sampled plants. If anything, publications resulting from that research group indicated that cities were able to support diverse plant communities (Aronson et al. 2014 Proc. R. Soc. B).

Figure 1: The boundary line for the City of LA would be helpful to use here to standardize with the remaining maps (Figs 3 & 4). Are major roads necessary to indicate here?

Figure 2: Since snails/slugs were the most urban tolerant taxonomic group, might it be beneficial to include at least one as an example here? Why were these three species selected beyond their urbanization ranks?

Line 323: Table 1 indicates that 487 species remained following the filters applied to the query. However, this statement states that 967 species remained following the filtered query and of those, 563 occurred more than once, 510 had UAI calculated, and 408 had UAI and occurred more than once. Where does n=487 fit in?

Table 1: For those unfamiliar with iNat, please indicate in the table title what ‘iNat Taxa ID’ represents (or can that information be moved to Supplemental Material?). I would also remind readers that these were specifically only native species of these taxonomic groups (i.e. were the non-native species removed pre- or post-filter?).

Table 2: This table might be better presented as a figure, with mean and range values indicated on a UAI gradient. Perhaps the UAI can be on the x-axis (centered at zero) with each taxonomic group on the y-axis, with the mean as a point and min/max UAI represented with lines radiating from the point (or perhaps smaller dots representing each species within the taxonomic group). This would provide a better visual of where each group lands regarding their urban tolerance/intolerance. If retaining these data as a table, ranges should at least be provided for each taxonomic group.

Line 358: Please define ‘IUM’.

Line 365: This final sentence is incomplete.

Discussion: It might be worth pointing out that some of the taxonomic groups studied were at the Family level, while others were at the level of taxonomic Order, which may obscure subtleties in urbanization responses. Even within one taxonomic Family/Order, there are large differences in urbanization response across native species and their populations (See Weiss et al. 2023 Ecology “Effect of species-level trait variation on urban exploitation in mammals”).

Reviewer #2: In this study the authors present a methodological approach on how to make use of open access opportunistic and citizen-science-based data to explore the relationship between the urban environment and biodiversity. The proposed methodology sounds reasoned and repeatable and gives insight into the association (or avoidance) of various species with the urban environment.

Below the authors can find some comments which I hope will help them to make some points of the manuscript clearer.

GENERAL COMMENTS

The discussion focuses a lot on the use of iNaturalist as a source of biodiversity data and tool for environmental city planning. However, although the statistical analysis included several taxa, the discussion only mentions the most tolerant and intolerant species (snails and slugs and some butterflies respectively), whereas it does not comment on the responses of the rest of the taxonomic groups. I think a paragraph describing the trends of the rest of the taxonomic groups could be added.

Although the end of the introduction section is rather untypical, as it summarizes methodology and includes some conclusions and suggestions, I would not disagree with this format if this was the authors’ choice.

The authors use the term ‘community science’ rather than ‘citizen science’ which is more commonly used in biodiversity studies. Similarly, they use the term ‘crowd-sourced data’. I suggest they also add “citizen science” either to the keywords or to the text to facilitate possible future search (e.g. for meta-analyses and review articles).

I suggest the authors avoid the word ‘urbanness’ and use more formal ones such as ‘urban intensity’, which is more widely used.

The family and order/class names should not be italicized according to the Zoological nomenclature (https://www.iczn.org/), only the species and the genera.

There are some abbreviations in the text which are not explained, e.g. PM2.5 (line 173), PM1.2b (line 136, 297), IUM (line 358).

SPECIFIC COMMENTS

L. 21 – Abstract: ‘city’ not capitalized.

L. 137: perhaps ‘repeated’ instead of ‘repeat’?

L. 174: What is the reference for PM2.5, Average Annual Traffic Volume, and Population Density? Although not used in the analysis due to collinearity, I suggest the authors add a reference for future use by other studies.

L. 198: Perhaps ‘selected’ instead of ‘select’?

L. 225-228: How many scientists from the Expert Council were asked to evaluate the data used? Did they evaluate all species included in this study (from the 12 taxonomic groups) one by one? Did all experts examine the same set of species?

L. 237-246: Have the authors considered the issue/effect of “iconic” and “charismatic” species that are more often recorded by citizens and amateurs? And alternatively, of species that are nearly never recorded because they are very common and abundant and observers think it is not worth noting them down?

L. 262: I think it would help to note here what is the number of 0.25 x 0.25 mile grid cells that were assessed.

L. 282: Thus, does this mean that each species occurred in at least 25 grid cells across the study region in all of the 11 sampling years?

L. 289-292: I think it would be helpful to add some more detail like “Species’ UAI scores are the logit-linear slope of the trend line ()” which is mentioned in line 339.

L. 298: Perhaps “of” instead of “or”?

L. 358: It is not defined what “IUM” is.

L. 365: It seems that the sentence is incomplete.

L. 373: I assume the authors refer to Figure 3 instead of 2.

L. 380: “a” is redundant.

L. 390: I think it would be helpful here to mention again what are the parameters measured for quantifying urban intensity (i.e. light pollution, impervious surfaces, noise pollution). Also, I suggest replacing ‘urbanness’ with ‘urban intensity’ (also l. 392).

Figures 3 & 4: Please make sure that the maps are colour-blind friendly.

Table 3: Do the authors think that it would be more informative to also add the CUTI values for Los Angeles city, together with the CUTI values for the entire study area?

6. PLOS authors have the option to publish the peer review history of their article (what does this mean?). If published, this will include your full peer review and any attached files.

Reviewer #1: **Yes: **Christine Rega-Brodsky

Reviewer #2: No

---

## [Author Response · Author response to Decision Letter 0]

12 Mar 2024

See attached document "ResponseToReviewers_PLOS_Curti.docx"

---

## [Editor Report · Decision Letter 1]

18 Mar 2024

Using Unstructured Crowd-sourced Data to Evaluate Urban Tolerance of Terrestrial Native Animal Species within a California Mega-City

PONE-D-23-39010R1

Dear Dr. Joseph Curti,

We’re pleased to inform you that your manuscript has been judged scientifically suitable for publication and will be formally accepted for publication once it meets all outstanding technical requirements.

Kind regards,

Dárius Pukenis Tubelis, Ph.D.

Academic Editor

PLOS ONE

Additional Editor Comments (optional):

Dear Dr Joseph Curti and Dr Morgan Tingley,

Thank you for submitting a corrected version of your manuscript PONE_D_23_39010.

Your responses to the suggestions/comments by reviewers were convincing. I also appreciated the changes that you have done along the manuscript.

With this, its quality was improved, and I now consider that your manuscript can be accepted for publication in PLOS ONE.

I made a final reading of the manuscript to check everything. I found a few minor problems that you have to fix before publication, maybe during the correction of proofs, or prior to it, when submitting the definitive files. Please find these corrections below.

PLOS ONE people migh contact you during the next days for details regarding figures, documents, etc...

Congratulations on your study!

Dárius P. Tubelis

PLOS ONE Editor

Additional things to fix:

Line 72. Fix with a space.

Line 182. Add a space before "and".

Line 197. Figure caption. It should be "Fig 1.". Abbreviated and without dot.

Lines 236-240. Maybe you can add "?" to these questions....

Line 274. Delete the comma after "groups".

Line 279. "recorded at least one record" sounds repetitive. Can you use "obtained" for the first one ?

Lines 326-330. Please check these numbers again. Just to make sure that they are correct.

Line 340. Figure caption. It should be "Fig 2.". Not dot and abbreviated.

Lines 365-366. Should orthoptera come without capital ? Not sure. It is the name of an Order.

Line 374. It should be "Fig 3.".

Line 379. The same....."Fig 4." then the caption.

Line 407. One bracket is in italics.

Line 417. The same.

Line 421. Replace "75," by a long dash.

Line 424. Add a space after "tool".

Line 442. Add a space before "but".

Line 443. The same before "Many".

Line 450. It should be Cooper et al. [81].

Line 454. Consider replacing by "They" to avoid repeating the name shown lines before.

Line 507. Delete ")".

Line 541. "city" ? no capital. Maybe you are wright. I do not know the rules there....sorry.

References

You are citing the DOIs in a wrong way. The correct is: https://doi.org/ then the numbers and letters. Please see recent papers if in doubt. You have to fix all.

Ref 5. Delete Editor.

Ref 7. The page numbers are missing.

Ref 14. Delete "Baselga A, Editor".

Ref 22. The pages are missing.

Ref 26. Delete Haddad, Editor.

Ref 28. Scientific name in italics.

Ref 35. It might be "Evol".

Ref 39. Is 277 correct ?

Ref 40. Abbreviate "Diversity".

Ref 64. Use long dash for pages.

Ref 67. Delete editor.

Ref 97. Is this citation of Ecography correct ?

Figure 2. You replaced "urbanness" along the text, so you have to do the same in this figure.

That is all.

Dárius

---

## [Editor Report · Acceptance letter]

3 May 2024

PONE-D-23-39010R1 

PLOS ONE

Dear Dr. Curti, 

I'm pleased to inform you that your manuscript has been deemed suitable for publication in PLOS ONE. Congratulations! Your manuscript is now being handed over to our production team.

Kind regards, 

on behalf of

Dr. Dárius Pukenis Tubelis 

Academic Editor

PLOS ONE